# Merging Multiphase CTA Images and Training Them Simultaneously with a Deep Learning Algorithm Could Improve the Efficacy of AI Models for Lateral Circulation Assessment in Ischemic Stroke

**DOI:** 10.3390/diagnostics12071562

**Published:** 2022-06-27

**Authors:** Jingjie Wang, Duo Tan, Jiayang Liu, Jiajing Wu, Fusen Huang, Hua Xiong, Tianyou Luo, Shanxiong Chen, Yongmei Li

**Affiliations:** 1Department of Radiology, The First Affiliated Hospital of Chongqing Medical University, No. 1 Youyi Road, Yuzhong District, Chongqing 400016, China; jingjiewang@126.com (J.W.); ljy1997lilian@163.com (J.L.); ltychy@sina.com (T.L.); 2College of Computer and Information Science, Southwest University, Tiansheng Road, Beibei District, Chongqing 400715, China; duooo7@163.com; 3Department of Radiology, NO. 958th Hospital of PLA Army, Chongqing 400020, China; wujiaj619@163.com; 4Department of Anesthesiology, The First Affiliated Hospital of Chongqing Medical University, Chongqing 400016, China; sammul2008@163.com; 5Department of Radiology, Chongqing General Hospital, University of Chinese Academy of Sciences, Chongqing 400013, China; rjdfxyh@163.com

**Keywords:** acute ischemic stroke, collateral circulation, large vessel occlusion, deep learning, 4D-CTA

## Abstract

We aimed to build a deep learning-based, objective, fast, and accurate collateral circulation assessment model. We included 92 patients who had suffered acute ischemic stroke (AIS) with large vessel occlusion in the anterior circulation in this study, following their admission to our hospital from June 2020 to August 2021. We analyzed their baseline whole-brain four-dimensional computed tomography angiography (4D-CTA)/CT perfusion. The images of the arterial, arteriovenous, venous, and late venous phases were extracted from 4D-CTA according to the perfusion time–density curve. The subtraction images of each phase were created by subtracting the non-contrast CT. Each patient was marked as having good or poor collateral circulation. Based on the ResNet34 classification network, we developed a single-image input and a multi-image input network for binary classification of collateral circulation. The training and test sets included 65 and 27 patients, respectively, and Monte Carlo cross-validation was employed for five iterations. The network performance was evaluated based on its precision, accuracy, recall, *F*_1_-score, and AUC. All the five performance indicators of the single-image input model were higher than those of the other model. The single-image input processing network, combining multiphase CTA images, can better classify AIS collateral circulation. This automated collateral assessment tool could help to streamline clinical workflows, and screen patients for reperfusion therapy.

## 1. Introduction

Stroke is the third leading cause of death and disability among Chinese residents, and acute ischemic stroke (AIS) accounts for 81.9% of all patients [1]. Acute large vessel occlusion (LVO) of the brain causes severe ischemia in the blood supply area and leads to severe disability and mortality. Good collateral circulation can maintain the basic metabolism of brain tissue in the penumbra region, slow down the expansion of infarct foci and disease progression, and improve the success rate of reperfusion [2]. Collateral circulation is crucial for the treatment and prognosis of patients with AIS [3,4]. However, assessing collateral circulation is a challenging topic due to the complexity and difficulty of quantifying the neurological vessels’ structure. Although previous studies have tried and reported various methods for assessing collateral circulation, there is no unified standard for assessing collateral circulation [5,6]. Moreover, even when the same assessment method is used, physicians have significant inter-rater differences. The key to saving stroke patients is to restore perfusion as early as possible [7,8,9,10,11], and the accurate assessment of collateral circulation is essential for patient treatment decisions.

CTA has become the primary assessment method of collateral circulation for AIS because it is fast and non-invasive. Since the establishment of the collateral circulation is a dynamic process, dynamic multi-phase CTA can show the details of collateral circulation, and some studies [12,13] have proved its advantages and clinical value. Four-dimensional computed tomography angiography (4D-CTA) is obtained while completing the emergency CT perfusion, so it has high temporal resolution and provides accurate images of the arterial, venous, and adjacent phases.

Deep learning, as a subset of artificial intelligence, can be applied to medical imaging and has shown promise for automated detection. Deep learning analyzes the task in a data-driven manner, and it automatically learns relevant data features from a specific data set. Through learning, the model selects the correct features from the training data and this enables the classifier to make the correct decisions when testing new data. The learning process is essentially an optimization problem-solving process.

The importance of the collateral circulation is well known, but due to the complexity of the collateral circulation assessment, pre-treatment evaluation inevitably causes a delay in reperfusion time. We hypothesized that adopting advanced deep learning algorithms to learn and model the information regarding the collateral circulation provided by dynamic CTA would provide a model for rapid, accurate and objective assessment of the collateral circulation; this could assist the clinic in obtaining a more detailed assessment of the condition prior to treatment to develop an individualized treatment plan.

## 2. Material and Methods

### 2.1. Patients

Ninety-two consecutive patients with AIS who were admitted to our stroke center from July 2020 to August 2021 were included in the study. The inclusion criteria were: (1) age ≥ 18 years; (2) complaints of acute neurological dysfunction located on one side of cerebral hemisphere; (3) an emergency one-stop whole-brain four-dimensional computed tomography angiography (4D-CTA)/CT perfusion (CTP) examination found occlusion of the internal carotid artery (ICA) or the M1/M2 segment of middle cerebral artery on one side and abnormal perfusion in the corresponding blood supply region; (4) standard treatment according to the guidelines for the early management of AIS patients; and (5) a complete medical record file of hospitalization. The Human Ethics Committee of the First Affiliated Hospital of Chongqing Medical University approved this study under approval number 2021-274.

The exclusion criteria were: (1) the presence of significant motion artifacts in the images that affect accurate observation and post-processing evaluation; (2) the combination of other neurological diseases such as cerebral hemorrhage and tumors; (3) the combination of vertebral-basilar system disease or posterior cranial fossa cerebrovascular disease; and (4) bilateral internal carotid artery or middle cerebral artery stenosis or occlusion.

Patient demographic and clinical data were collected, including age, sex, time from onset to examination, National Institutes of Health Stroke Scale Baseline Score (NIHSS) at time of admission, TOAST typing of acute stroke etiology, treatment protocol, and whether hemorrhagic transformation occurred.

### 2.2. CT Examination Equipment and Scanning Protocols

A 320-row detector CT scanner (Aquilion Vision, Canon Medical Systems Corporation, Otawara, Japan) was used for all patients’ whole-brain one-stop 4D-CTA/CTP examination. The patient was placed in the supine position, and the patient’s head was immobilized using a head brace and straps to reduce motion artifacts. The scan was performed from the greater foramen magnum to the cranial vault. Iodine 400 (Iopamidol 400, Bracco Sine, Italy) was used as the contrast agent. The scans and contrast injection were started simultaneously, and after 7 s of contrast injection, the dynamic volume perfusion scan was started. First, the non-contrast computed tomography (NCCT) was performed in dynamic volume single scan mode, followed by a 4D-CTA/CTP scan in whole-brain dynamic volume interrupted acquisition mode. Data were acquired in 2 s intermittent scans during the arterial phase (11–36 s) and 5 s intermittent scans during the venous phase (40–60 s). The total acquisition time was 60 s, and 19 volumes were acquired, including NCCT, CTP, and 4D-CTA. The scanning parameters were: 80 kV, 150–310 mA, spherical tube speed 0.75 s, coverage 140–160 mm, field of view (FOV) 240 mm × 240 mm, matrix 512 × 512, an adaptive iterative reconstruction algorithm was used to reduce the dose, a reconstruction layer with a thickness of 1.0 mm and layer spacing of 1.0 mm was applied to improve the reconstruction speed. The total dose was 5.0–6.0 mSv (k = 0.0021). The contrast protocol was performed using high-pressure injector P3T technology (MEDRAD Stellant CT Injection System, Bayer Medical Care, Pittsburgh, USA), which automatically calculates the contrast dose and rate based on patient gender, weight, height, and contrast concentration.

### 2.3. Image Preprocessing

All image data obtained from the whole brain one-stop 4D-CTA/CTP scans, totaling 19 volumetric data packages, were selected and imported into a workstation (Vitrea, fX, 1.0, Canon Medical Systems Corporation, Japan) for post-processing. The data were loaded into the “SVD+ algorithm-based deconvolution method for cerebral perfusion” protocol, and the software system automatically labeled the inflow arteries and outflow veins for post-processing. After the automatic post-processing was completed, the operator manually checked whether the arterial and venous curves were selected accurately. The requirements were as follows: the peaks of the arterial and venous time curves were prominent and noticeable, and were single-peaked “bell-shaped,” with no double or multiple peaks; the arterial and venous starting position was after 0 s on the time axis, and ended after the outflow platform period; the peak of the arterial curve was in front, and the peak of the venous curve was behind. If the temporal density curve had double or multiple peaks, motion artifact calibration was first performed; secondly, arterial and venous points were re-selected. The arteries with the earliest enhancement were generally selected, e.g., ICA and MCA. Regarding the selection of veins, the superior sagittal and transverse sinuses were usually set. Recalculation was performed after the choice had been made and the images were generated. 

The image was reconstructed at a layer thickness of 1 mm and spacing of 1 mm. According to the arterial–venous time–density curve, the computed tomography angiography (CTA) of the peak phase of the arterial curve was defined as the “arterial phase”, the CTA of the intersection of the arterial and venous curves was defined as the “arterial–venous phase”, and the computed tomography venous angiography (CTV) of the peak phase of the venous curve was defined as the “venous phase”. The CTV of the first phase after the venous curve dropped into the platform phase was the “late venous phase”. The CTA/CTV of each phase was reconstructed, then the data from NCCT was used to perform subtraction and bone removal to obtain the corresponding “arterial phase”, “arterial–venous phase”, “venous phase”, and “venous late phase” volume packet after vascular subtraction.

The subtracted CTA/CTV-MIP maps were obtained from the subtracted volume data of each CTA/CTV by the maximum density projection (MIP) algorithm. The CTA/CTV-MIP maps of four phases (arterial phase, arteriovenous phase, venous phase, and late venous phase) were stitched together (Figure 1). The MIP algorithm was implemented by Python 3.8.0, and the packages we used were opencv(4.5.3) and PIL(8.0.1). The DICOM images were converted into pixel values by normalization according to CT values, and the sequence images of each period were converted into an array of size N × 512 × 512, N was the number of mono-temporal sequence images, and the values of the points (x, y) on the maximum density projection map were assigned to the maximum pixel values along the z-axis.

### 2.4. Collateral Circulation Grading

Collateral circulation grading was performed using a modified ASITN/SIR (American Society of Interventional and Therapeutic Neuroradiology/Society of Interventional Radiology) collateral grading scale based on dynamic multi-period CTA [5,14,15]. The scale consists of 5 levels: level 0 indicates no or few collateral branches in the ischemic area at any stage; level 1 is partial collateral circulation until the late venous phase; level 2 is partial collateral circulation in the ischemic area before the venous phase; level 3 is complete collateral circulation formation in the ischemic area in the late venous phase, and level 4 is total collateral circulation before the venous phase. Grades 0 to 2 were defined as poor collateral circulation (Figure 2), while grades 3 to 4 were defined as good collateral circulation (Figure 3). We marked patients as having good or poor collateral circulation according to each patient’s dynamic CTA collateral grading. Two experienced neuroradiologists performed the scoring, and they negotiated a solution if they had a different opinion. The scoring process was blinded to the clinical data.

### 2.5. Model Construction and Training

We tried the commonly used classification networks, VGG, Densenet, ResNet34, etc., and finally chose the ResNet34 as it had the best classification effect. The Adam optimizer was selected during the training process, its learning rate was set to 0.0001, and the batch size was set to 16. The training was stopped beyond ten epochs or when the loss value stopped decreasing to prevent overfitting. We also used migration learning to take the weights obtained from pre-training on ImageNet and assign them to the initialization weights so that the model can eventually distinguish between good and poor collateral circulation. The framework for model-building neural networks was Pytorch (1.7.0), and we also used package pandas (1.1.3) to create datasets. All experiments were performed under 64-bit Windows OS, and the network was trained on NVIDIA GeForce RTX 2070 SUPER.

The multi-period stitched image that had undergone image pre-processing first passed through a 3 × 3 convolution layer, then a 3 × 3 maximum pooling layer, and then a 4-layer residual layer, where the residual layer consisted of residual blocks (Figure 4), and the residual block consisted of two 3 × 3 convolution layers and a batch normalization layer. The feature maps of the input residual blocks were added element by element to the output feature maps, with the first layer containing three residual blocks, the second layer containing four residual blocks, the third layer containing six residual blocks, and the fourth layer containing three residual blocks. The final fully-connected layer was used as a classifier, mapping the output features into two categories of actual number distributions, and the softmax layer mapped the two real numbers into two (0–1) category probability values, and the sum of the two category probability values was 1. The category with the higher probability value was the final prediction class (Figure 5).

The multi-image input processing network was consistent with the structure of the classification network used in the single-input image model. The difference was that the images of the four periods were input into four independent network branches, and the four branches corresponding to each category outputted four probability values. The four probability values were averaged to obtain the final category probability values, with the type with the higher probability value being the last prediction class (Figure 6).

The demographic and clinical characteristics of all patients in the groups with good and poor collateral circulation were statistically described and compared. Continuous variables conforming to normal distribution were presented as mean ± standard deviation, the two-independent-samples *t*-test was used to compare two groups, and one-way ANOVA was used to compare multiple groups. Continuous variables that did not conform to a normal distribution were recorded using the median (inter-quartile spacing), comparisons between two groups were made using the Mann–Whitney U test, and comparisons between multiple groups were made using the Kruskal–Wallis test or one-way ANOVA. Count variables were recorded as the number of patients or percentages, and the χ^2^ test was used to compare groups.

Five Monte Carlo cross-validations were applied to each of the two models. The ratio of the training and validation sets was 70% and 30%. Specifically, each training included 65 patients in the training set and 27 patients in the validation set. The performance indicators of the models were accuracy, precision, recall, *F*_1_-score, and area under the subject’s working characteristic curve (AUC).

Accuracy: the probability of correct judgment in the total sample size.
$$\mathrm{Accuracy}\;=\frac{tp+tn}{tp+tn+fp+fn}$$

Precision (positive predictive value): the probability of a true positive in a sample judged to be positive.
$$\mathrm{Precision}\;=\frac{tp}{tp+fp}$$

Recall (sensitivity): the probability that a sample of actual true positives was predicted to be positive.
$$\mathrm{Recall}\;=\frac{tp}{tp+fn}$$

*F*_1_-score: Defined as the harmonic mean between precision and recall, and used to evaluate the effectiveness of the binary classification model. (*F*_1_-score values range from 0 to 1, where 1 represents the best output of the model and 0 illustrates the worst output of the model.)
$$F_{1}-score=2\cdot \frac{\;\mathrm{precision}\;\cdot \;\mathrm{recall}\;}{\;\mathrm{precision}\;+\;\mathrm{recall}\;}$$

## 3. Results

Ninety-two patients who suffered from AIS with large vessel occlusion were included in this study. The baseline clinical characteristics of the patients were as follows: ① 61 were male, and 31 were female, with an average age of 66.24 ± 13.64 years; ② distribution of onset time: 32 patients within 6 h, 19 patients within 6–24 h, and 41 patients over 24 h; ③ responsible vessels: 64 patients had a middle cerebral artery (MCA) occlusion (with or without anterior cerebral artery (ACA) occlusion), 14 patients had an internal carotid artery (ICA) occlusion, 13 patients had ICA and MCA occlusion, and there was 1 case of ICA, MCA, and ACA occlusion; ④ the median NIHSS score was 8.50 (12); ⑤ the modified ASITN/SIR collateral score based on dynamic multi-period CTA was 3 (1) points; ⑥ TOAST typing included 63 patients of large artery atherosclerosis type (LAA), 24 patients of cardiogenic embolism type (CE), and five patients of stroke of other determined etiology (SOE); ⑦ treatment methods included 22 patients who had endovascular intervention, eight patients had thrombolytic treatment, three patients had thrombolytic bridging and endovascular intervention, and 59 patients were subjected to conservative treatment. ⑧ There were 19 patients with hemorrhagic transformation.

Patients were divided into two groups based on good or poor collateral circulation, and statistically compared with each other regarding the above indicators (see Table 1 for details). Comparing the two groups with good collateral circulation and poor collateral circulation, showed that age, baseline NIHSS score, occluded artery, hemorrhagic transformation, and dynamic CTA ASTIN/SIR collateral classification were statistically different between the two groups (all *p* < 0.05). Gender, onset time, treatment method, and TOAST typing showed no statistical difference between the two groups (*p >* 0.05).

The training time for the single-image input processing network was 69 min, the prediction time was 0.26 s, the accuracy was 0.852 ± 0.045, the precision was 0.932 ± 0.034, the recall was 0.827 ± 0.076, and the *F*_1_-score was 0.860 ± 0.044. The training time for the multi-image input processing network was 120 min, the prediction time was 0.37 s, the accuracy was 0.822 ± 0.017, precision was 0.571 ± 0.081, recall was 0.813 ± 0.056, and *F*_1_-score was 0.836 ± 0.008 (Table 2). The single-image input processing network outperformed the multi-image input processing network in all indexes.

ROC curves were plotted for single-image and multi-image input networks for dichotomous collateral circulation evaluation (Figure 7 and Figure 8). The dashed line represents the ROC curve for each Monte Carlo cross-validation. The solid line is the average ROC curve for five iterations. The gray-shaded area indicates the interval of ±1 standard deviation. The AUC is labeled in each figure. The average AUC of the single-image input processing network for evaluating the collateral circulation was 0.89 ± 0.05, which was far more accurate than the multi-image input model.

## 4. Discussion

A study has shown that reperfusion within the first 100 min after emergency CT imaging helps to reduce mortality in patients with AIS [16]; therefore, it is essential to establish a reliable and fast assessment method for grading collateral circulation to help optimize reperfusion therapy patient selection. Our study developed two new collateral circulation evaluation networks, namely, a single-image input processing network and multi-image input processing network, based on the ResNet34 classification network. The single-image input processing network could perform a dichotomous assessment of collateral circulation with higher accuracy, precision, recall, FI score, and AUC. 

The main advantages of our model are: first, four-phase dynamic CTA/CTV data for collateral circulation evaluation used in this study was extracted from a whole-brain one-stop 4D-CTA/CTP examination. Frolich et al. [17] demonstrated that the 4D-CTA technique could better describe the collateral circulation, which could be due to the establishment of the collateral circulation as a dynamic process. Moreover, 4D-CTA/CTP provides more accurate images of the arterial, arteriovenous, venous, and late venous phases based on perfusion time–density curves, and does not additionally increase the radiation dose. This multi-phase dynamic CTA evaluation of collateral flow has a high temporal resolution and a smoother baseline between patients, providing reliable original data for collateral circulation evaluation, enabling a complete assessment of collateral circulation [18] and improving diagnostic accuracy [19], as well as providing value in determining clinical prognosis [12]. Second, a residual convolutional neural network [20] was employed, which can improve the performance of image classification tasks while deepening the network. Deep convolutional neural networks perform well in image classification tasks, and increasing the network depth can directly improve the network feature extraction, but it creates the problem of gradient loss. We used the ResNet classification network by combining the residual blocks, and upper-level features were passed to the bottom through cross-layer connections. The features of different layers could be given to each other, which further improved the network’s performance and alleviated the gradient loss problem. Third, the pre-processing method was simple, and saved time and labor costs. The Vitrea post-processing workstation used in this study had a robust and rigid alignment subtraction technique, which maximized the removal of skull and brain tissue effects, gave a clean background to the MIP images, and achieved similar results in similar studies that used very complex data pre-processing methods.

There are few similar studies. Ryan et al. [21] also performed an automated grading study of the collateral circulation, in which they used single-phase arterial CTA subtraction images and pre-processed reconstructed axial MIP, coronal MIP, axial MIP + coronal MIP, and 3D images as inputs to build a model for automated assessment of the collateral circulation. They found that the network model with axial MIP as input had the highest AUC value (of 0.93 ± 0.01). Therefore, this study selected the axial MIP images as model input. CTA/CTV subtraction images of the artery, arteriovenous, venous, and late venous phase were added as modeling input. The input images were processed in two ways, the first one stitched the four stages of CTA/CTV subtraction images of each subject into one image for model learning, and the second input the four stages of CTA/CTV subtraction images of each subject into the four branches of the model separately for learning, and finally, the average probability was calculated. The performance of the single-image input processing model was significantly higher than that of the multi-image input processing model, and the average AUC value was slightly lower (0.89 ± 0.05) than that of Ryan’s study. However, the accuracy and recall were higher than that of Ryan’s model, while overall, the two models had similar efficacy. The reason for the significantly greater AUC and accuracy of the single-image input processing model compared to the multi-image input processing model may be due to the practice of stitching the four-phase CTA/CTV subtraction images into a single image input model for learning, which allows the computer to more clearly learn the changes in the vessel volume of the same patient over different phases, i.e., increasing the information about the collateral circulation at different phases and reducing the model training time, thus improving the learning efficiency and classification performance.

Mumu et al. [22] performed an automated collateral circulation evaluation using 4D-CTA images. They firstly established a 4D-CTA cerebrovascular standard template using data from 12 normal subjects, then they made unfilled vessel subtraction images after aligning the patient’s data with the established cerebrovascular template, and performed a trichotomous analysis of poor, moderate, and good collateral circulation based on the latter. Finally, they used 4D-CTA data from 46 patients with AIS for testing and obtained a collateral circulation classification model with an AUC of 0.85. This approach avoided the extensive work of collecting data and training the model, and allowed us to make an intuitive judgment regarding poorly filled vessels. The study had three shortcomings: first, there are many variants in the human cerebral vessels. For example, the literature reports five variants in the anterior circulation and ten variants in the posterior circulation of the Willis circle in normal subjects [23], the probability of such variants was more pronounced in the population of patients with AIS [24], the branching variants of the distal vessels are more complex, and atherosclerotic type stroke patients have significantly tortuous and elongated large arteries. These conditions lead to difficulties in aligning the patient’s CTA images with the template established with healthy human CTA data, and modeling the assessment of patients with fit human data will cause significant systematic errors. Secondly, the critical advantage of 4D-CTA is that its high temporal resolution can show the blood velocity alterations during the establishment of the collateral circulation in stroke patients. Our results showed that the normal side of the cerebral vessels in patients with LVO-AIS also suffered from significant blood velocity alterations, which were characterized by a marked delay in filling of the draining veins and even a failure to empty the contrast on the affected side at the end of the 4D-CTA scan, which is quite different from normal subjects. Third, blood circulation also varies in each individual, and even with the same scanning modality, the velocity of vascular visualization can vary between patients. It is impossible to obtain accurate results on this basis by using each phase of 4D-CTA to align and compare the differences in cerebral vascular volumes. Therefore, although all images of 4D-CTA were applied in this study, we believe that the results may not be as accurate as comparisons between specific phases that are filtered based on perfusion curves.

This study also found that (i) patients with good collateral circulation were younger than those in the other group, as reported in previous studies [25], which may be related to the greater circulatory compensatory capacity of younger patients; (ii) patients with good collateral circulation had lower NIHSS scores, consistent with previous studies, as patients with good collateral circulation tended to have smaller infarct cores and slow core growth rates, and therefore had milder disease and lower stroke scores [26]; and (iii) patients with good collateral circulation had a lower incidence of hemorrhagic transformation, consistent with previous studies [2], again demonstrating the protective effect of collateral circulation.

There are some limitations of this study that need to be considered: (1) we only performed a dichotomous evaluation of collateral circulation. The modified ASITN/SIR collateral grading scale based on dynamic multi-period CTA consists of 5 levels, Grades 0 to 2 were defined as poor collateral circulation, while grades 3 to 4 were defined as good collateral circulation according to previous studies. Dichotomizing simplified the statistical analysis and led to the easy interpretation and presentation of results. Indeed, dichotomizing may lead to several problems such as loss of information and reduced statistical power, but the loss of information is quite small compared to the five-group category. Furthermore, the present sample size was relatively small for a deep learning algorithm. Therefore, we believe that dichotomizing the 5-level collateral circulation grading as a classification criterion for computer deep learning models produced more reliable results under the current sample size. We will further explore the use of deep learning algorithms for multi-category evaluation of collateral circulation in a future study with a larger data set; (2) this study is a single-center study, and all 4D-CTA images were acquired with the same imaging equipment. Different equipment parameters, contrast dosage, and subtraction methods may create differences in the original images. Future multi-center studies should be conducted to promote network applicability further; and (3) the number of patients included in the study was relatively small. Patients with large vessel occlusion are relatively few in acute ischemic stroke; thus, the sample size was not large enough. However, we have used cross-validation to address this issue.

## 5. Conclusions

The single-image input processing network, which included stitching multi-period CTA images as input, can better classify AIS collateral circulation. This automated collateral assessment tool can help streamline clinical workflows, and screen patients for reperfusion therapy.

## Figures and Tables

**Figure 1 diagnostics-12-01562-f001:**
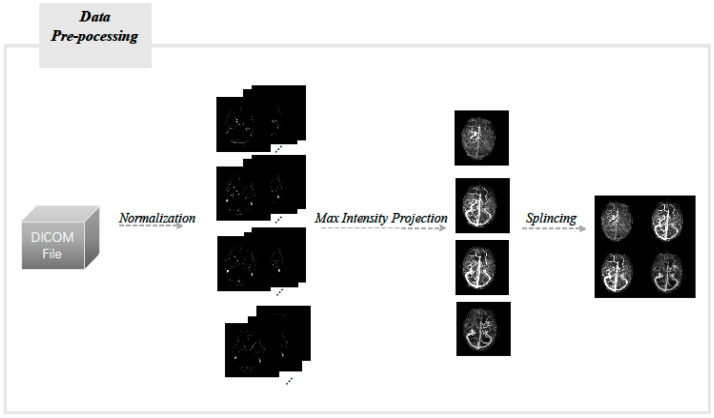
Flow chart of image pre-processing.

**Figure 2 diagnostics-12-01562-f002:**
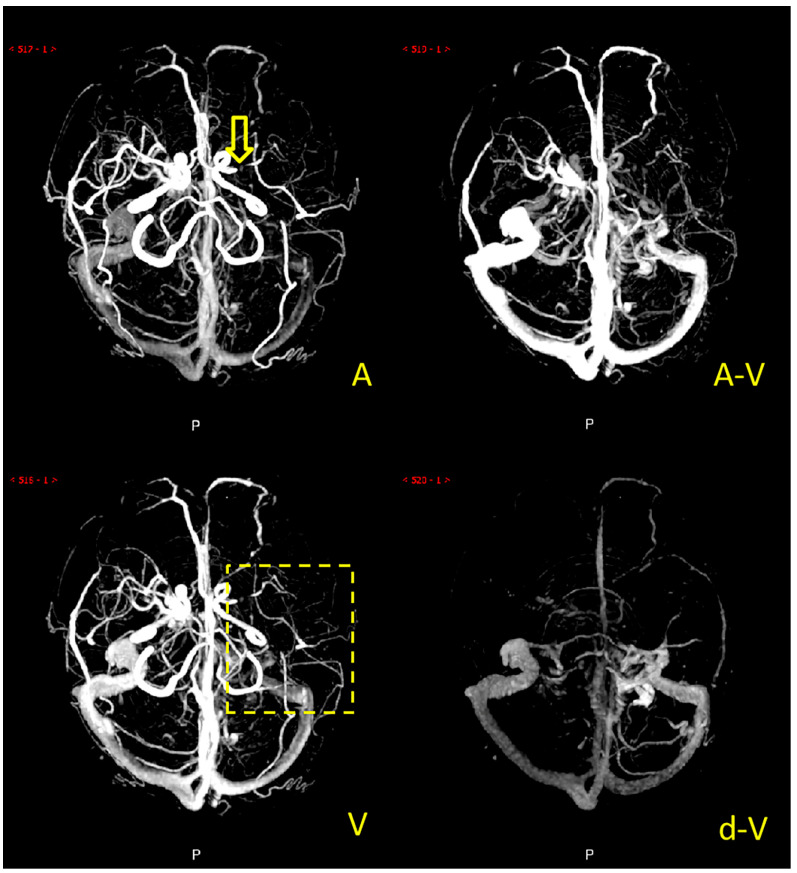
Example of a patient with poor collateral circulation. Note: The patient’s left middle cerebral artery was occluded (the yellow arrow), and there was no major collateral vessel in the left middle cerebral artery blood supply area (the yellow dashed box) of A (arterial phase), A-V (arteriovenous phase), V (venous phase), and d-V (venous late phase).

**Figure 3 diagnostics-12-01562-f003:**
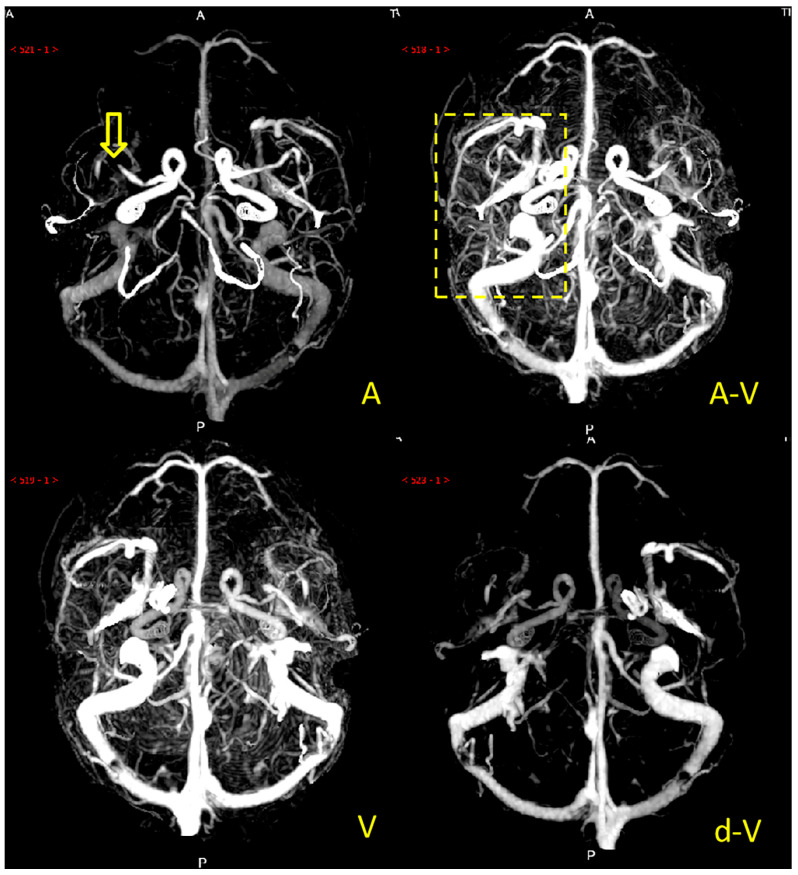
Example of a patient with good collateral circulation. Note: the M1 segment of the right middle cerebral artery of the patient was occluded (the yellow arrow), and there were many collateral vessels in the blood supply area (the yellow dashed box) of the right middle cerebral artery in A (arterial phase), A-V (arteriovenous phase), V (venous phase), and d-V (venous late phase).

**Figure 4 diagnostics-12-01562-f004:**
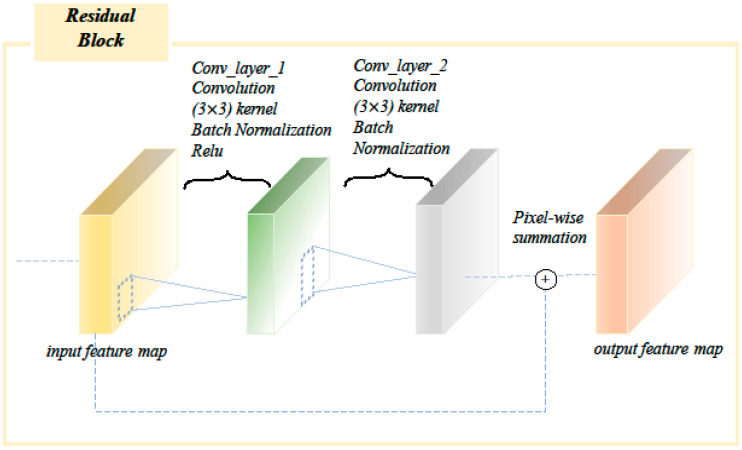
Residual block structure in the network. The residual block consisted of two 3 × 3 convolution layers and a batch normalization layer.

**Figure 5 diagnostics-12-01562-f005:**
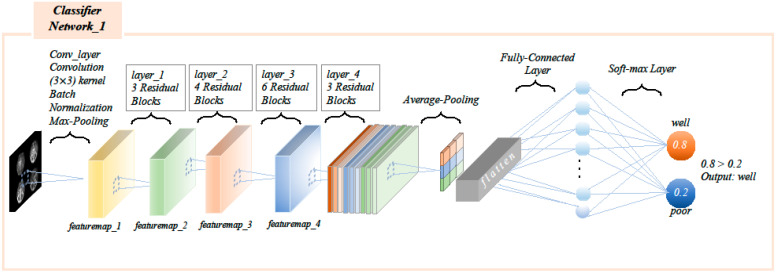
Single image input processing network. The feature maps of the input residual blocks were added element by element to the output feature maps, with the first layer containing three residual blocks, the second layer containing four residual blocks, the third layer containing six residual blocks, and the fourth layer containing three residual blocks. The final fully-connected layer was used as a classifier, mapping the output features into two categories of actual number distributions, and the softmax layer mapped the two real numbers into two (0–1) category probability values, and the sum of the two category probability values was 1. The category with the higher probability value was the final prediction class.

**Figure 6 diagnostics-12-01562-f006:**
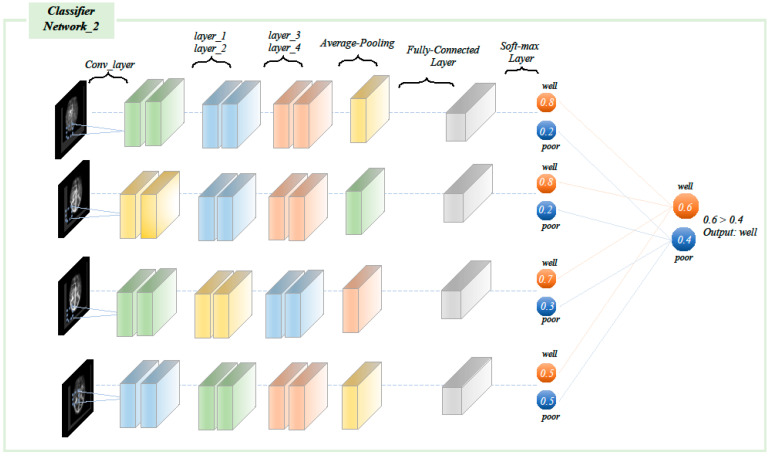
Multi-image input processing network statistical analysis and model testing. The images of each different phase were separately input into the network of the same structure, each branch produced a classification result, and the final result was calculated by the weighted average.

**Figure 7 diagnostics-12-01562-f007:**
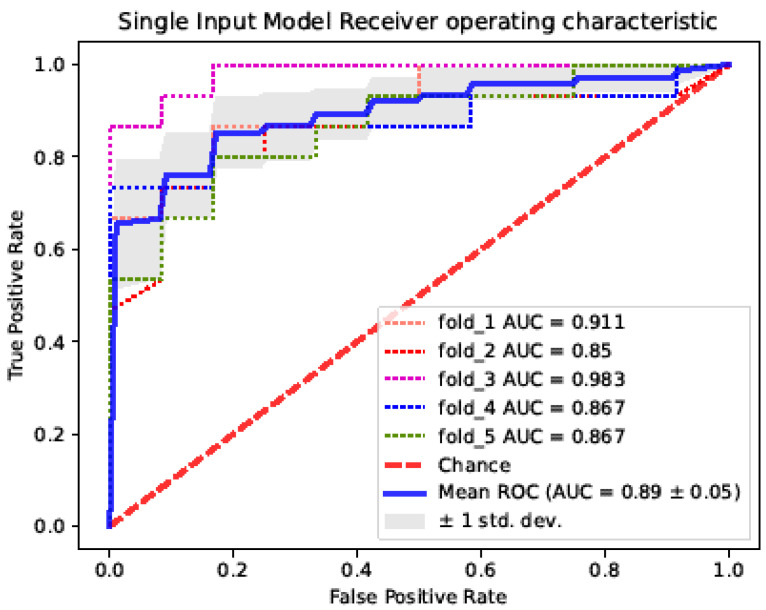
ROC curve of the single-image input processing network collateral circulation evaluation model.

**Figure 8 diagnostics-12-01562-f008:**
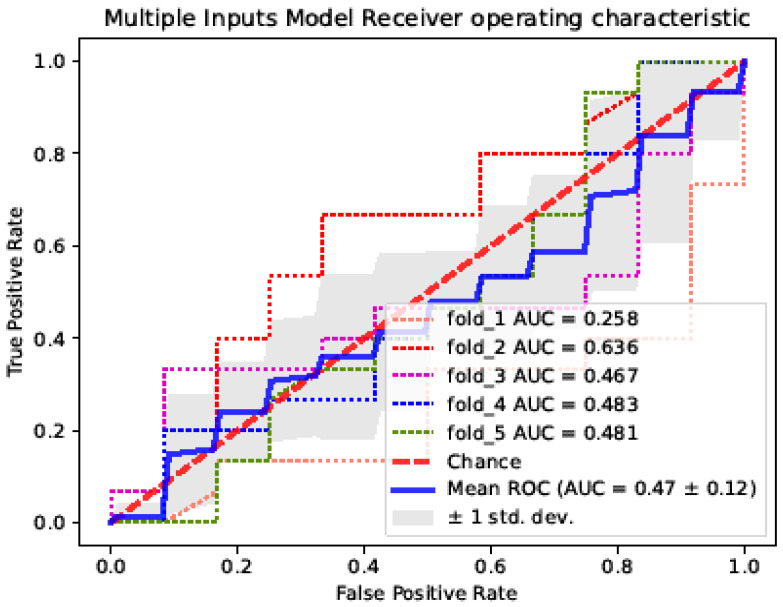
ROC curve of the multi-image input processing network collateral circulation evaluation model.

**Table 1 diagnostics-12-01562-t001:** The description and comparison of clinical characteristics between groups with good or poor collateral circulation.

	All Patients	Poor Collateral	Good Collateral	*p*-Value
**Gender (Male/%)**	61/66	30/71	31/62	0.341
**age**	66.24 ± 13.64	69.45 ± 14.05	63.54 ± 12.82	**0.038**
**NIHSS,M(IQR)**	8.50 (12)	12 (10)	3.50 (9)	**0.001**
**Onset time**				0.132
**<6 h**	32	17	15	
**6–24 h**	19	11	8	
**>24 h**	41	14	27	
**Treatment method**				0.729
**Thrombolysis**	8	5	3	
**Endovascular interventions**	22	9	13	
**Thrombolytic bridging intervention**	3	1	2	
**Conservative treatment**	59	27	32	
**Responsible Artery**				**0.041**
**ICA**	14	3	9	
**MCA (ACA)**	64	28	36	
**ICA + MCA**	13	10	3	
**ICA + MCA + ACA**	1	1	0	
**TOAST type**				0.126
**LAA**	63	25	38	
**CE**	24	15	9	
**SOE**	5	2	3	
**Hemorrhage conversion** **(cases/%)**	19/21	13/31	6/12	**0.025**
**collateral grading**	3 (3)	0 (2)	4 (1)	**<0.001**

Note: Age, NIHSS, occluded artery, hemorrhagic transformation, and dynamic CTA ASTIN/SIR collateral grading were statistically different between groups with good and poor collateral circulation (all *p* < 0.05).

**Table 2 diagnostics-12-01562-t002:** Performance of single-image input processing network and multi-image input processing network.

Network	Accuracy	Precision	Recall	*F*_1_-Score
**Single-image input**	0.852 ± 0.045	0.932 ± 0.034	0.827 ± 0.076	0.860 ± 0.044
**Multi-image input**	0.822 ± 0.017	0.571 ± 0.081	0.813 ± 0.056	0.836 ± 0.008

Note: The single-image input processing network outperforms the multi-image input processing network on all indicators, and the differences in AUC and accuracy rate were especially obvious.

## Data Availability

The data presented in this study are available on request from the corresponding author. The data are not publicly available due to privacy or ethical considerations.

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
