# Peer review of "Merging Multiphase CTA Images and Training Them Simultaneously with a Deep Learning Algorithm Could Improve the Efficacy of AI Models for Lateral Circulation Assessment in Ischemic Stroke"

_diagnostics, 2022, doi:10.3390/diagnostics12071562_

Round 1
Reviewer 1 Report
line 55: “scholars”: rather “in the previous studies, …”
58: “subjective differences” do you mean “inter-rater differences” ?
74: please capitalize the W from “we”
77: would you highlight there what it can bring for clinical practice (reduce inter-operator variability)
81: were “included in the study. Inclusion criteria were” instead of “collected…included” ?
175: Figure 2, please add an arrow and asterisk to show the site of the occlusion and the collateral circulation on the images themselves
176: instead of “obvious” please use for example “major”
206: Figure 4, writing is very small, is challenging to read; Would it be possible to further explain in Figure legend ? idem for the other Figures, in particular Figure 6
383: “Limitations of this study: (1) only a dichotomous evaluation of collateral circulation was performed” would it be possible to develop there ? In the Discussion section it would be interesting that the authors comment on the choice of a dichotomous classification.
Author Response
Thank you for all your suggestions and time spent on the manuscript, we have revised the manuscript based on your suggestions and the detailed responses to the questions have been uploaded as an attachment. Thanks again.

Reviewer 2 Report
The paper is well written and interesting. My main complaint is that the English of the manuscript can be somewhat spotty. Beyond that, the manuscript is well structures, methodologically very sound and the results are interesting and applicable in future work.
My comments are as follow:
1. English of the manuscript should be improved. For example I assume "The image was reconstructed as a layer thickness of 1 mm and spacing of 1 mm." should be written as "The image was reconstructed at a layer thickness of 1 mm and a spacing of 1 mm.". There are many examples as this one where English proficiency affects the clarity of the writing.
2. Line 152: Python should be capitalized. If you are listing the programming language, you should also list the libraries used for processing (my assumption is PyDICOM/Numpy/PIL) along with their versions.
3. "Dicom" should be written as "DICOM".
4. Figures are very low resolution - especially figures 5 and 6, but noticeable in figures 7 and 8 as well - they should be redrawn or saved at a higher DPI if possible.
5. When discussing the limitations of the study, please note the relatively low amount of patients in the study - 92. While this is not a methodological error, as you have taken steps to address it (cross-validation), it should be noted as one of the possible limitations, as it is possible certain statistical variations simply were not included.
Kindest regards,
the reviewer
Author Response
Thank you for all your suggestions and time spent on the manuscript, we have revised the manuscript based on your suggestions and the detailed responses to the questions have been uploaded as an attachment. Thanks again.

This manuscript is a resubmission of an earlier submission. The following is a list of the peer review reports and author responses from that submission.